# Reading Behaviors through Patterns of Finger-Tracking in Italian Children with Autism Spectrum Disorder

**DOI:** 10.3390/brainsci12101316

**Published:** 2022-09-29

**Authors:** Claudia Marzi, Antonio Narzisi, Annarita Milone, Gabriele Masi, Vito Pirrelli

**Affiliations:** 1Institute for Computational Linguistics, National Research Council, 56124 Pisa, Italy; 2IRCCS Stella Maris Foundation, 56018 Pisa, Italy

**Keywords:** reading, autism, finger-tracking, developing readers, prediction-driven processing

## Abstract

The paper proposes an ecological and portable protocol for the large-scale collection of reading data in high-functioning autism spectrum disorder (ASD) children based on recording the finger movements of a subject reading a text displayed on a tablet touchscreen. By capitalizing on recent evidence that movements of a finger that points to a scene or text during visual exploration or reading may approximate eye fixations, we focus on recognition of written content and function words, pace of reading, and accuracy in reading comprehension. The analysis showed significant differences between typically developing and ASD children, with the latter group exhibiting greater variation in levels of reading ability, slower developmental pace in reading speed, less accurate comprehension, greater dependency on word length and word frequency, less significant prediction-based processing, as well as a monotonous, steady reading pace with reduced attention to weak punctuation. Finger-tracking patterns provides evidence that ASD readers may fail to integrate single word processing into major syntactic structures and lends support to the hypothesis of an impaired use of contextual information to predict upcoming stimuli, suggesting that difficulties in perception may arise as difficulties in prediction.

## 1. Introduction

Autism spectrum disorder (ASD) is characterized by persistent deficits in social communication and interaction, along with the presence of restrictive and repetitive behaviors [1]. The incidence of ASD is growing worldwide, with rates up to 1/44 in the USA, according to the Centers for Disease Control and Prevention (CDC) [2], and 1/87 in Italy [3]. With ASD being a heterogeneous clinical condition, disentangling specific phenotypes according to measurable dimensions may be helpful for more tailored diagnosis and interventions [4].

In recent years, the use of technology to improve the phenotyping of people with ASD has made considerable progress, especially in the area of eye micromovements [5,6]. In particular, the synergistic interaction of eye and hand movements in the exploration of a visual scene displayed on a touchscreen device was shown to provide a congruent signature of the “attention maps” of subjects with ASD and control individuals [7]. A familiar context of visual and tactile interaction is when children use the finger of their dominant hand to point to the letters of written words during early acquisition of their reading skills [8,9]. In previous work [10,11], a common tablet and a web application [12] were used to record the finger movements of Italian developing readers when they concurrently read and finger-point to a text displayed on a tablet touchscreen. Text comprehension was also controlled by asking children to answer a few multiple-choice questions on the text content after each reading session. Despite the undoubtedly different dynamics of ocular and tactile movements [13], finger movements [10] were shown to replicate established reading effects in a language with a transparent orthography such as Italian [14]. Namely, shorter tracking times were reported for higher-frequency words, shorter words, and words in denser neighborhoods, with a significant interaction with years of schooling. In particular, facilitation effects were reported to become smaller after grade 3, suggesting that normally developing children end up memorizing larger orthographic lexicons, i.e., lexicons that integrate increasingly longer and rarer words, even in a language (Italian) whose transparent orthography does not require a lexical reading strategy [15,16].

Here, we report on patterns of finger-tracking of Italian early readers with ASD and compare them with the same behavior in typically developing (TD) children, sampled according to their real distribution in grade-matched primary school classes (cluster sampling). The aim of this preliminary study is to determine whether an analysis of finger-tracking patterns in reading Italian can be useful to accurately differentiate reading skills of children with ASD from those of a control group of typically developing children. With this purpose in mind, we used a tablet and a web application to acquire the finger patterns of ASD children reading an Italian connected text and compared them with the finger patterns of age- and grade-matched TD children reading the same text. The patterns were analyzed focusing on three complementary aspects of our finger-tracking data set: pace of word reading, pace of reading of multi-word intonation units, and text comprehension. These aspects allowed us to focus on different component skills that are assumed to underpin the ability to read fluently: (i) efficient orthographic lexical access, supporting top-down control of decoding and online word-level comprehension, (ii) use of both explicit and implicit grammatical prosody in aloud and silent reading, and (iii) text-level reading comprehension. As to point (i), a process-oriented analysis of written word recognition in reading a connected text was intended to integrate and complement recent evidence of a developmental difficulty in word reading by children with ASD [17] (in spite of their strength in alphabetic or syllabic decoding [18,19,20]), which appears to be in contrast with previous literature [21,22]. In connection with point (ii), the implicit prosody hypothesis [23,24,25] claims that silent readers activate prosodic patterns as induced by punctuation marks such as commas and periods, which then influence the way readers process and access text content at the sentence level and beyond. Some studies comparing high-functioning ASD participants to matched TDs reported deficits in the ability to perceive grammatical prosodic cues, while others did not identify such a group difference [26]. By measuring tracking time latencies at sentence and phrase boundaries, we intended to assess ASD readers’ sensitivity to the implicit prosody encoded in written texts through punctuation marks [27] and their ability to process multi-word intonational units [26]. Finally, by directly tapping into readers’ text-level comprehension (in connection with point (iii)), we expected to establish a causal link between online reading skills, i.e. the ability to decode and integrate local written information during text processing, and the ability to develop an understanding of the text as a whole [28].

Based on our present analysis of the complex, dynamic integration of reading skills in children with ASD, we suggest that the use of a tablet as a tactile interface for visual perception analysis can offer a robust experimental protocol for large scale data screening in ASD research and clinical settings. The proposed method is based on a friendly, ecological and versatile application which proves to shed light on intricate patterns of tracking behavior as potential non biological markers for identification of children with ASD through the task of reading a connected text, both silently and aloud.

## 2. Materials and Methods

### 2.1. Sample

87 children were included in the study. The sample was composed of 20 children (19 m, 1 f) with a diagnosis of ASD (age range 6:10–10:11 years) attending primary school classes, from the 2nd to the 5th grade, and 67 TD (age range 7:1–11:2, with a balanced gender distribution). Participants with ASD were recruited at the Department of Child Psychiatry and Psychopharmacology of IRCCS Stella Maris of Calambrone (Pisa). The clinical diagnosis of ASD was established according to DSM-5 criteria [1] and confirmed using algorithm cutoffs on the ADOS-2 [29], which was administered by an ADOS research reliable examiner (A.N.). Intelligence Quotient (IQ), Verbal Comprehension Index (VCI) and Perceptual Reasoning Index (PRI), evaluated through WISC-IV, was greater than 95. Detailed figures of sample distribution are given in Table 1 and Table 2.

Exclusion criteria were as follows: (a) neurological syndromes or focal neurological signs; (b) significant sensory impairment; (c) anamnesis of birth asphyxia, premature birth, head injury or epilepsy; (d) use of any psychotropic medication; and (e) potential secondary causes of ASD determined by high-resolution karyotyping, DNA analysis of Fragile-X or screening tests for inborn errors of metabolism. TD children were sampled in a primary school in the city of Pisa (Italy) from a population of pupils who matched the ASD group for age and grade levels (a Welch two-sample *t-test* was performed on age distributions in the two experimental groups by grade level) and were reported to have neither special education needs nor cognitive disorders. A written informed consent was obtained from the parents of all participants. The study was approved by the Ethics Committee of the National Research Council of Italy (approval number: CNR 0037523/2021) and was conducted according to the ethical standards of the Declaration of Helsinki and the Italian Association of Psychology (AIP).

### 2.2. Procedures

The two samples were assessed according to patterns of finger-tracking in connection with three complementary aspects of reading behavior: 1. Recognition of both content and function words; 2. The pace of reading multi-word intonation units; 3. Text comprehension (controlled with multiple-choice questions on text content). An ICT platform and a common tablet front-end [12] were used for the online monitoring of silent and aloud reading abilities in early graders. Children were asked to read a connected text displayed on a common tablet (10.1 inches Samsung TAB A SM-T510N, 1.8 GHz Octa-Core, 3 GB RAM, 64 GB eMMC, Android 10, with a 14.9 cm × 24.5 cm screen with a resolution of 1920 × 1200 pixels), written in Arial font, size 21.25pt. Readers were instructed to finger-point to each word in the text while reading it, as is common practice with beginning readers.

During reading, the tablet produced an online record of the sliding movements of the finger captured by the touchscreen, together with a sound track of the voice of the reader, recorded in aloud reading sessions through the tablet built-in microphone. The reading text was broken down into episodes. At the end of each text episode, children were asked to answer a written questionnaire with multiple choice answers to control for their reading comprehension. The ICT platform (developed within the *Readlet* project, www.readlet.it) captured a multimodal stream of time-aligned signals, including voice recording (in aloud reading), patterns of finger-pointing to text during reading (both silent and aloud), and the reader’s answers to content questions as well as the time taken to answer each question. Data were transmitted to a centralized server for post-processing, where finger-tracking time series were aligned with the text and analyzed accordingly. The tablet could thus map a continuous sliding movement on the touchscreen into a discrete series of densely distributed touch events, and a convolution algorithm precisely reconstructed where on the screen the reader’s finger was pointing to at each reading time tick. In the end, a series of discretized touch events was mapped onto text lines, allowing one to know the letter pointed to by the reader’s finger at any time during reading. Ultimately, we took the finger-tracking time of each text unit as a proxy of the processing time taken by a child while reading the unit (see [13] for a preliminary comparison of finger-tracking and eye-tracking evidence in adult reading).

### 2.3. Materials

Children were presented with two fantasy narrative texts, which they had to read either silently or aloud. In both reading conditions, children were instructed to finger-track the text they were reading on the tablet screen. Texts were preliminary annotated on several levels of linguistic analysis to automatically assess their linguistic difficulty/complexity. Each of the four texts was divided into 5 episodes of increasing complexity. Two *wh*-questions with multiple-choice answers were presented to the readers at the end of each episode, for verifying their ability to retrieve the general content of the story and identify temporal or cause-effect relations.

Descriptive statistics of administered texts are given in Table 3.

### 2.4. Data Analysis

We analyzed text-aligned time series of finger-tracking data with *R* [30]. Statistical modeling was conducted with regression mixed models (i.e., generalized additive models, or *GAMs*), and results were displayed with the help of either linear regression plots or distribution plots. GAMs with fixed and random predictors have been preferred over linear mixed models since the former do not necessarily assume an a priori linear relation between covariating features. This is particularly useful in a developmental perspective (for a detailed description of GAM regression models, see [31]). Results are detailed in Section 3.

## 3. Results

By way of preliminary analysis, we controlled for the finger tracking time of each word in the reading texts for both reading modalities (aloud and silent) in the two experimental groups (see per-word tracking time distributions in Figure 1). ASD children showed greater variation in tracking time than TD children, especially in the aloud reading condition, and their word tracking time was significantly longer than controls’ in both reading modalities (*p* < 0.001).

To ensure the comparability of our data patterns with established behavioral evidence, we checked if we could replicate basic word length and word frequency effects on reading ([14,32,33]), using word finger-tracking duration as a proxy of word eye-fixation duration. As expected, finger tracking time was found to correlate positively with word length and negatively with word frequency in both groups and reading modalities (Figure 2). In addition, ASD children were found to be more affected by word length and word frequency effects than TD children and exhibited a more pronounced time difference between silent and aloud reading. In fact, for increasing length and decreasing frequency, words take significantly more time to be read by ASD children than TD children (*p* < 0.001), as shown in the regression plots of Figure 2. The pertinent regression models are reported in Table 4.

### 3.1. A Developmental Perspective

In a developmental perspective, TD children showed a progressive decrease in both word length and word frequency effects on their reading times, with 2nd graders, who take considerable advantage from short and highly frequent words, being overall more sensitive to such effects than older readers. The facilitatory effect of lexical frequency and the inhibitory effect of word length on tracking time of word tokens are reported for the four grade levels in Figure 3.

The pertinent regression models are reported in Table 5.

The behavior of ASD readers did not replicate exactly the same developmental pattern of typical readers, as shown in Figure 4.

The pertinent regression models are reported in Table 6.

On average, irrespective of their attended grade level, ASD children were more sensitive to word length and word frequency than TD children (*p* < 0.001 for both length and frequency.

We ran additional *GAM* models predicting tracking time with either length or (log) frequency as independent variable in interaction with reading modality and the experimental groups. The explained deviance for the two *GAMs* is, respectively, R^2^ 38.2% and R^2^ 29.9%, where for increasing levels of word length and (log) frequency, we found corresponding increasing model slope values (respectively, positive slope for length and negative slope for log frequency), as shown in Figure 5. Steeper model slopes for ASD vs. TD children confirm the higher sensitivity of the former to word length and word frequency effects.

In addition, ASD children showed a reduced developmental effect compared to TDs, with a less prominent reduction of word length and frequency effects for increasing grade levels.

### 3.2. Word Internal Processing

At this stage of analysis, we shifted our focus to the intra-word serial dynamic of finger tracking to take into account the predictive nature of word reading. To analyze speed changes within a word, we modeled the tracking time for each individual letter making up a written word token and contrasted content words with function words. Figure 6 shows the decreasing tracking time for each word symbol (letter) when moving from the onset of a word towards its end in TD children. Interestingly, function words were predicted more quickly during recognition than content words. This behavior is consistent with length and frequency distributions of content versus function words, with the former being attested in a 1–15 length range (mean =6.12) and a 1–15.67 (log) frequency range (mean =9.53), and the latter in a 1–10 length range (mean =2.67) and a 1–15.65 (log)frequency range (mean =13.54).

Related regression models are reported in Table 7.

The same pattern is observed in ASD children, with a significantly smaller difference between content and function words and a significantly less prominent bias to predict words during recognition, as shown in Figure 7.

For ASD children, there is no significant difference in symbol tracking time between content and function words (*p* = 0.38), as opposed to what we observe in TD children (*p* < 0.005).

Related regression models are reported in Table 8.

Both experimental groups tend to accelerate their reading pace when moving from the beginning of each word (onset) toward its ending, and the effect is more pronounced in tracking function words as opposed to content words. This behavioral pattern suggests that readers in the 7–11 age range start developing the ability to anticipate the end of a written word as soon as they recognize it. However, the development of this ability is not as apparent in ASD children as it is in TD children, at least in this age range.

### 3.3. Text Comprehension

As reported in Section 2, we asked our participants to answer a few multiple-choice questions on text content after each text episode in order to control for text comprehension. Distributions of comprehension accuracy scores are reported in Figure 8.

We observe an incremental accuracy rate for increasing grade levels in the TD group versus a greater variability in the ASD group. This is particularly evident in the silent reading condition, where we suspect that some ASD children might have not read the text carefully enough. Comprehension may have been affected by the genre of the texts we used in our experimental protocol, namely fantasy narrative texts, which may have not raised the interest and attention of our target ASD children. Undoubtedly, the silent reading modality requires an additional attention load, which may challenge the performance of ASD subjects. There is consensus that attention difficulties may be a characteristic feature of ASD (for a systematic review, see [34]).

Accordingly, we found more untracked word tokens in the ASD than in the TD group: 17.28% in ASD as compared to 14.38% in TD, especially in silent reading (57.21%, with 42.79% in aloud reading). See details as reported in Table 9.

### 3.4. Prosody and Chunking

Finally, we focused our analysis on both phrase-level and sentence-level punctuation marks to measure the latency between the tracking offset of a written word that immediately precedes a punctuation mark and the tracking onset of the immediately ensuing word. A longer latency provides an indication of a stronger perception of syntactic boundaries and a better online processing of multi-word units. Conversely, a shorter latency provides evidence for a more fragmented processing strategy during reading. To distinguish intra-sentence processing effects from inter-sentence processing effects, we separately analyzed latency at weak punctuation marks (e.g., commas) and latency at strong punctuation marks (e.g., periods).

Figure 9 shows the different distributions of latency times at weak punctuation marks across our experimental groups in both aloud and silent reading. Here, latency times are normalized by the readers’ tracking pace to take into account the difference in tracking speed between ASD and TD readers (see Figure 1). Normalized latency at weak punctuation marks appears to be significantly lower in ASD children than in TD children.

It is worth reporting that, whereas weak punctuation latency is significantly different between the two groups in both reading modalities (*p* < 0.001), strong punctuation latency is not (*p* = 0.16 for aloud reading, and *p* = 0.30 for silent reading).

## 4. Discussion

In summary, the present analysis of finger-tracking patterns confirms the high heterogeneity in levels of reading ability of ASD children, with some readers showing clear evidence of difficulty decoding (Figure 1) and impaired reading comprehension (Figure 8). This is in line with the consensus view that ASD children tend to show reading comprehension deficits, e.g., [21,35,36], and it accords well with evidence that difficult reading comprehension is also a common weakness in high-functioning ASD children [37].

It has been suggested that reading comprehension in ASD readers may fail due to a weakness in maintaining and updating the semantic knowledge required to build up the mental model of a running text [38]. This suggestion is often coupled with the observation that comprehension may be challenged by an impairment of the ability to integrate the meaning of single linguistic units (e.g., words or chunks) into a holistic, high-level semantic representation. This latter aspect appears to distinguish ASD subjects from typical ones [39], thus affecting their reading comprehension profiles. These suggestions are coherent with the well-known autism *Theory of Mind* [40], which proposes an impaired ability in inferring and understanding represented or described thoughts and actions in ASD subjects. It is worth reiterating here that in our data sample, comprehension may have been affected by the particular genre of texts we used in the experimental protocol, namely fantasy narrative texts, which may not have raised the interest and attention of our target ASD children.

Our experimental evidence shows that there is a consistent pattern in per-word tracking times for comprehension accuracy ranges in both experimental groups: despite higher tracking times and a reduced accuracy in comprehension in ASD children, tracking time tends to be significantly (*p* < 0.001) reduced for increasing accuracy levels in text comprehension in both experimental groups. We interpret this evidence as indicative of a possible connection between reading comprehension and the online processing strategies adopted by ASD readers. Most reading models agree that fluent readers adopt a mixed strategy combining efficient recognition of orthographic words as holistic units (i.e., the lexical reading route) and knowledge of how to map sub-lexical strings onto syllables (i.e., the sub-lexical reading route) [41,42]. Especially when confronted with a highly transparent orthographic system such as the Italian one, developing readers are expected to be more prone to following a sub-lexical route initially [43] and then successfully integrate sub-lexical and lexical reading strategies as they become more proficient. Our experimental results for the TD group are in good accord with the evidence reported for Italian children in the literature [14,32,43]. In Italian, good levels of reading accuracy are reached quite rapidly by TD children, with also difficult, slow readers making comparatively few decoding errors. It has been shown that word length affects the reading speed of early graders, with its effect decreasing considerably at later grades for high-frequency words. Additionally, the word frequency effect has been shown to reach its full potency in 3rd grade, when children start memorizing orthographic representations for long, low-frequency words [14].

Despite a great heterogeneity, our ASD group showed, as a whole, a slower developmental pace in lexical reading and a weaker bias towards predictive processing (Figure 4, Figure 6 and Figure 7). This evidence provides an indication that ASD developing readers find it more difficult to integrate a bottom-up alphabetic/syllabic reading strategy with a top-down, lexically driven strategy, whereby words are read after their lexical orthographic representations are accessed in the mental lexicon, rather than before they are recognized. In addition, our evidence of a significantly lower normalized latency at weak punctuation marks in ASD readers than in TD ones (Figure 9) suggests that the former adopt a more fragmented and local parsing strategy, with shorter pauses at multi-word chunk boundaries. We observed, in fact, a significant difference in weak punctuation latency between the two groups in both reading modalities (*p* < 0.001), whereas we did not find a significant difference in strong punctuation latency between the two groups (*p* = 0.16 for aloud reading and *p* = 0.30 for silent reading). This suggests that although the sentence represents a natural reading unit for ASD readers, smaller syntactic units are likely to be neglected. It is important to emphasize that our evidence is limited to the processing of implicit prosody during reading and does not contradict evidence that ASD subjects can effectively perceive and use prosodic clues for syntactic chunking in natural speech processing [44,45].

Further analyses are needed to verify whether such a persisting bottom-up, local reading strategy is determined by an impoverished orthographic lexicon (as is the case with many difficult developing readers) or is rather the result of a non-predictive processing bias that is more specific to ASD readers, or a combination thereof. Our results lend support to the hypothesis of an impaired anticipatory use of contextual information to predict upcoming stimuli during a reading task, be they words or units larger than words, such as phrasal chunks and intonation units. Atypicality in perception has been suggested to be a key feature of ASD as a consequence of a reduced integration of experience [46,47]: this makes it more difficult to use knowledge derived from the past to generate predictions about the occurrence of new events [48]. Our data seem to also suggest that, in a complex cognitive activity such as reading, perception difficulties in ASD children may be coupled with difficulties in prediction. Predictive processing is a fundamental component in language comprehension, both in sentence comprehension during word-by-word reading [49] and intra-word processing [50,51,52]. In fact, it has been shown that efficient language processing also relies on generating successful anticipation about upcoming input, and that word predictability influences word reading times (see, among many others, [33,53,54,55]). We conjecture that a reduced prediction-driven mechanism, in a serial processing task such as reading, may represent a distinctive feature of the way ASD children process a written text, and that this feature is in line with the comprehensive cognitive profile of ASD. Accordingly, ASD children tend to read words thoroughly to their completion. This evidence, coupled with a different developmental pattern in connection with word length and word frequency effects, may suggest that ASD children tend to rely more heavily on a sublexical reading route and to a lesser extent on a lexical reading route. In the end, an exceedingly reduced reliance on predictive symbolic processing can be conjectured to lie at the roots of a wide range of reading patterns in ASD children, from difficulty decoding to poor reading comprehension.

## 5. Conclusions

Our preliminary findings confirm the heterogeneous nature of reading skills in children with ASD. Although much more data should be collected, including by varying the genre of texts submitted to participants, our results show a fairly coherent behavioral pattern in early ASD readers: a slower developmental pace in lexical reading, combined with a weaker bias towards predictive processing. In fact, on a less local level of linguistic analysis, the two observations may have a common explanation. The tracking pace of ASD readers provides some evidence that they may find it hard to integrate processing units incrementally, e.g., by combining syllables into words and words into major syntactic structures. This was confirmed by their relative neglect of weak punctuation marks and direct speech turns, an effect concomitant with a flat prosodic intonation of their oral reading. It is remarkable that it was possible to collect such a wealth of evidence using a simple tablet screen as a tactile interface for visual perception analysis. We believe that this tool can offer a robust experimental protocol for the large-scale, multimodal collection of naturalistic data for an extensive assessment and longitudinal monitoring of the reading behavior of ASD children.

## Figures and Tables

**Figure 1 brainsci-12-01316-f001:**
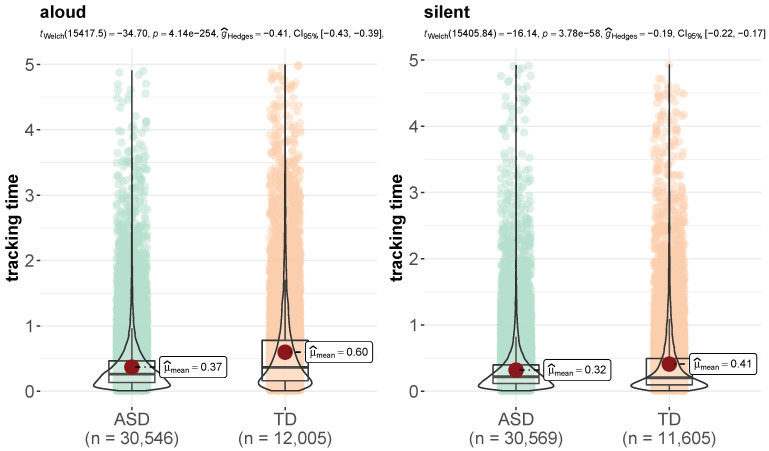
Distributions of word tracking time (in seconds) in ASD and TD children in the aloud (**left**) and silent (**right**) reading modality (plot obtained with the *ggplot2* package, function *ggbetweenstats*).

**Figure 2 brainsci-12-01316-f002:**
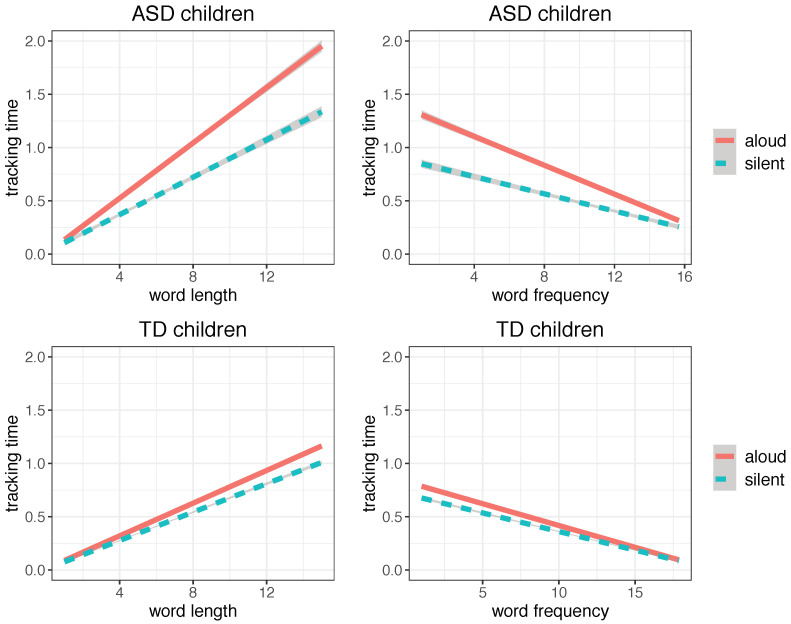
Linear regression plots for word length and word frequency in ASD (**top**) and TD children (**bottom**) in the aloud and silent reading modalities, fitting word tracking time (in seconds) (plot obtained with the *ggplot2* package, function *lm*; shaded areas indicate 95% confidence interval).

**Figure 3 brainsci-12-01316-f003:**
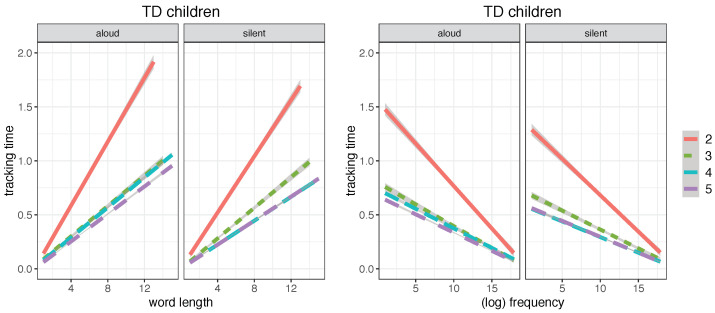
Linear regression plots for word length (**left**) and word frequency (**right**) in TD children for the four grade levels (from 2nd to 5th) in the aloud and silent reading modalities, fitting word tracking time (in seconds) (plot obtained with the *ggplot2* package, function *lm*; shaded areas indicate 95% confidence interval).

**Figure 4 brainsci-12-01316-f004:**
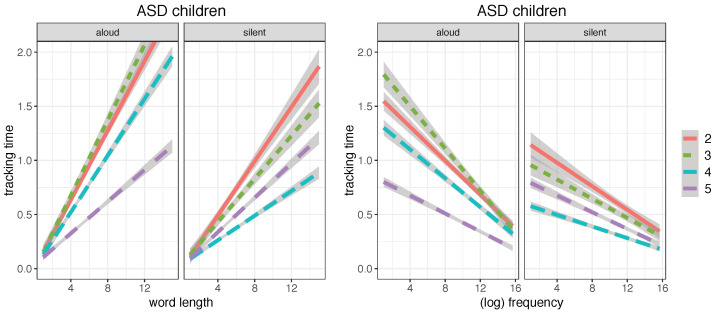
Linear regression plots for word length (**left**) and word frequency (**right**) in ASD children, for the four grade levels (from 2nd to 5th) in the aloud and silent reading modalities, fitting word tracking time (in seconds) (plot obtained with the *ggplot2* package, function *lm*; shaded areas indicate 95% confidence interval).

**Figure 5 brainsci-12-01316-f005:**
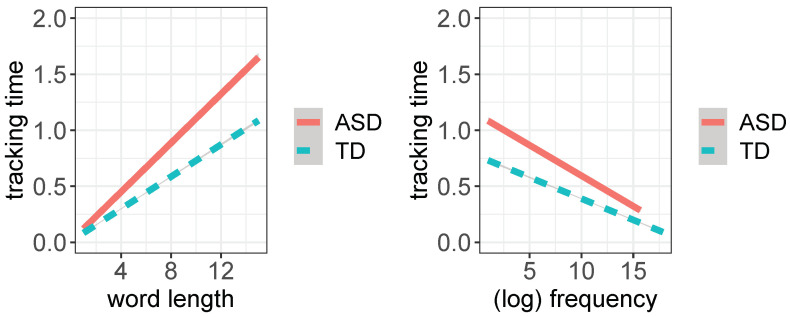
Linear regression plot for word length (**left**) and word frequency (**right**) for both ASD and TD children (plot obtained with the *ggplot2* package, function *lm*).

**Figure 6 brainsci-12-01316-f006:**
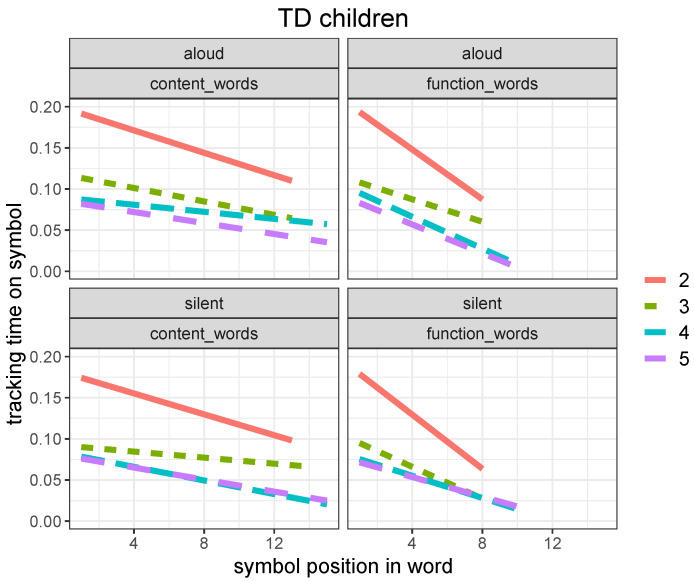
Linear regression plot for symbol tracking time (in seconds) as a function of their position within the word (with word onset for x=1), for aloud (**top**) and silent (**bottom**) reading in TD children for the four grade levels (from 2nd to 5th) for content (**left**) versus function (**right**) words (plot obtained with the *ggplot2* package, function *lm*).

**Figure 7 brainsci-12-01316-f007:**
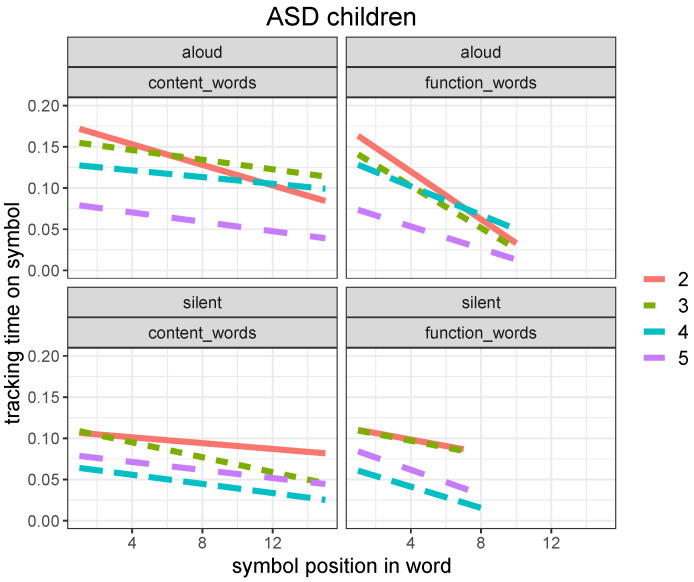
Linear regression plot for symbol tracking time (in seconds) as a function of their position within the word (with word onset for x=1) for aloud (**top**) and silent (**bottom**) reading in ASD children for the four grade levels (from 2nd to 5th) for content (**left**) versus function (**right**) words (plot obtained with the *ggplot2* package, function *lm*).

**Figure 8 brainsci-12-01316-f008:**
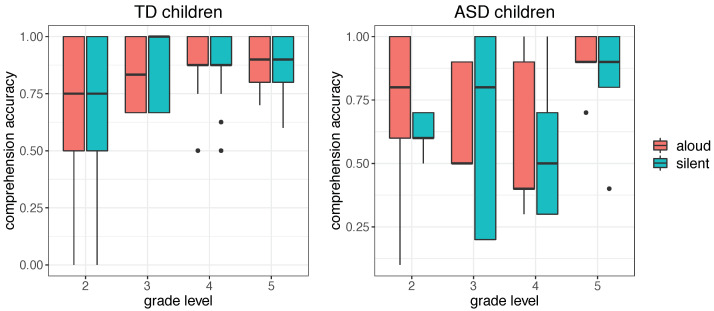
Boxplot distributions of accuracy scores (normalized in a 0–1 range) for aloud and silent reading in TD and ASD children for the four grade levels (from 2nd to 5th).

**Figure 9 brainsci-12-01316-f009:**
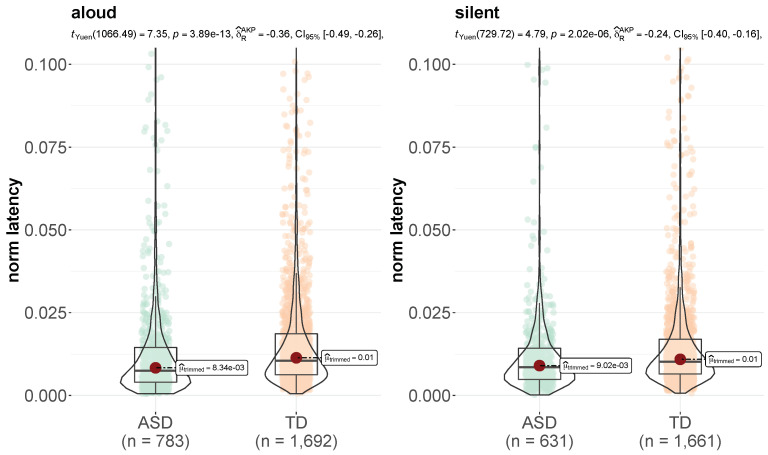
Distributions of normalized latency time for weak punctuation in ASD and TD children in the aloud (**left**) and silent (**right**) reading modality (plot obtained with the *ggplot2* packages, function *ggbetweenstats*).

**Table 1 brainsci-12-01316-t001:** Sample details for each group per grade levels.

ASD	Gender	Age Range	Mean (sd)
2nd grade	5 m	6.10–8.5	7.8 (0.10)
3rd grade	4 m	8.8–9.4	9.08 (0.3)
4th grade	4 m	9.4–10.9	10.2 (0.9)
5th grade	6 m; 1 f	10.5–10.11	10.5 (0.3)
**TD**	**Gender**	**Age Range**	**Mean (sd)**
2nd grade	12 m; 11 f	7.1–8.4	7.8 (0.5)
3rd grade	2 m; 5 f	8.6–9.1	8.10 (0.2)
4th grade	10 m; 10 f	9.3–10.6	9.9 (0.5)
5th grade	10 m; 7 f	10.2–11.2	10.7 (0.5)

**Table 2 brainsci-12-01316-t002:** Clinical and demographic features of participants per group.

	N	Age Range	Gender	ADOS-2 Comp Score	IQ
ASD	20	6.10–10.11	19 m; 1 f	6.3±1.4	104.2±10.3
TD	67	7.1–11.2	34 m; 33 f	-	-

**Table 3 brainsci-12-01316-t003:** Detailed statistics for experimental input per episode averaged over the 4 narrative texts.

Text Episode	N Tokens: Mean (sd)	Token Length: Range	Token Length: Mean (sd)	Content Word logfreq: Mean (sd)	Funct. Word logfreq: Mean (sd)
1st episode	157.26 (1.92)	1–10	4.21 (2.17)	10.42 (2.54)	14.05 (1.96)
2nd episode	173.76 (1.64)	1–13	4.66 (2.46)	9.67 (2.89)	13.19 (3.69)
3rd episode	187.32 (4.46)	1–14	4.71 (2.53)	9.59 (2.90)	13.65 (2.74)
4th episode	188.09 (5.88)	1–15	4.87 (2.76)	9.50 (2.92)	13.27 (3.76)
5th episode	201.12 (9.89)	1–15	5.07 (3.01)	9.31 (3.00)	13.37 (3.49)

**Table 4 brainsci-12-01316-t004:** *GAM* coefficients for models fitting tracking time as a function of word length (*length*) or (log)frequency (*freq*) and reading modalities for the two experimental groups. In the two models, participants are added as random effects (*re*).

ASD Group	Estimate	SE	t Value	Pr (>|t|)
intercept (aloud)	−0.01	0.06	−0.09	0.92
length (aloud)	0.13	0.01	58.35	<$2\times 10^{-16}$
length (silent)	−0.04	0.01	−13.48	<$2\times 10^{-16}$
participants (re)	<$2\times 10^{-16}$		R^2^	30.1%
intercept (aloud)	1.36	0.06	21.94	<$2\times 10^{-16}$
freq (aloud)	−0.07	0.01	−39.49	<$2\times 10^{-16}$
freq (silent)	0.03	0.01	10.91	<$2\times 10^{-16}$
participants (re)	<$2\times 10^{-16}$		R^2^	22.7%
**TD Group**	**Estimate**	**st.err**	**t Value**	**Pr(>|t|)**
intercept (aloud)	0.08	0.04	1.99	<0.05
length (aloud)	0.07	0.01	111.58	<$2\times 10^{-16}$
length (silent)	−0.01	0.01	−10.64	<$2\times 10^{-16}$
participants (re)	<$2\times 10^{-16}$		R^2^	42.9%
intercept (aloud)	0.89	0.04	23.44	<$2\times 10^{-16}$
freq (aloud)	−0.04	0.01	−73.05	<$2\times 10^{-16}$
freq (silent)	0.01	0.01	7.31	<0.001
participants (re)	<$2\times 10^{-16}$		R^2^	32.7%

**Table 5 brainsci-12-01316-t005:** *GAM* coefficients for models fitting tracking time as a function of word length (*length*) or (log)frequency (*freq*) in interaction with grade levels (2, 3, 4, 5) for the two reading modalities. Participants in the control group (TD) are added as random effects (*re*).

TD Group	Estimate	SE	t Value	Pr(>|t|)
intercept aloud 2	0.01	0.06	0.08	0.94
length (aloud 2)	0.15	0.01	80.77	<$2\times 10^{-16}$
length (aloud 3)	−0.08	0.01	−25.03	<$2\times 10^{-16}$
length (aloud 4)	−0.08	0.01	−36.42	<$2\times 10^{-16}$
length (aloud 5)	−0.08	0.01	−40.27	<$2\times 10^{-16}$
length (silent 2)	−0.02	0.01	−6.65	<0.001
length (silent 3)	0.02	0.01	4.30	<0.001
length (silent 4)	0.01	0.01	0.71	0.48
length (silent 5)	0.01	0.01	2.82	<0.005
participants (re)	<$2\times 10^{-16}$		R^2^	45.8%
intercept aloud 2	1.54	0.06	24.89	<$2\times 10^{-16}$
freq (aloud 2)	−0.08	0.01	−52.48	<$2\times 10^{-16}$
freq (aloud 3)	0.04	0.01	14.89	<$2\times 10^{-16}$
freq (aloud 4)	0.04	0.01	23.91	<$2\times 10^{-16}$
freq (aloud 5)	0.04	0.01	25.75	<$2\times 10^{-16}$
freq (silent 2)	0.01	0.01	3.76	<0.001
freq (silent 3)	−0.01	0.01	−0.75	0.45
freq (silent 4)	−0.01	0.01	−0.11	0.91
freq (silent 5)	0.01	0.01	−1.15	0.25
participants (re)	<$2\times 10^{-16}$		R^2^	34.3%

**Table 6 brainsci-12-01316-t006:** *GAM* coefficients for models fitting tracking time as a function of word length (*length*) or (log)frequency (*freq*) in interaction with grade levels (2, 3, 4, 5), for the two reading modalities. Participants in the ASD group are added as random effects (*re*).

ASD Group	Estimate	SE	t Value	Pr (>|t|)
intercept aloud 2	−0.01	0.12	−0.08	0.94
length (aloud 2)	0.16	0.01	37.71	<$2\times 10^{-16}$
length (aloud 3)	0.01	0.01	2.06	<0.05
length (aloud 4)	−0.03	0.01	−5.00	<0.001
length (aloud 5)	−0.09	0.01	−14.72	<$2\times 10^{-16}$
length (silent 2)	−0.03	0.01	−5.39	<0.001
length (silent 3)	−0.04	0.01	−4.09	<0.001
length (silent 4)	−0.04	0.01	−4.32	<0.001
length (silent 5)	0.04	0.01	4.71	<0.001
participants (re)	<$2\times 10^{-16}$		R^2^	32.2%
intercept aloud 2	1.59	0.13	12.72	<$2\times 10^{-16}$
freq (aloud 2)	−0.08	0.01	−23.66	<$2\times 10^{-16}$
freq (aloud 3)	−0.02	0.01	−3.56	<0.001
freq (aloud 4)	0.01	0.01	1.62	0.11
freq (aloud 5)	0.03	0.01	7.69	<0.001
freq (silent 2)	0.02	0.01	3.73	<0.001
freq (silent 3)	0.03	0.01	4.06	<0.001
freq (silent 4)	0.02	0.01	3.63	<0.001
freq (silent 5)	−0.02	0.01	−2.38	<0.05
participants (re)	<$2\times 10^{-16}$		R^2^	24.1%

**Table 7 brainsci-12-01316-t007:** *GAM* coefficients for models fitting tracking time on symbols as a function of symbol position in the word (*position*) in interactions with content vs. function words (*cont, func*) and grade level (*class*) for the two reading modalities. Participants in the control group (TD) and word tokens are added as random effects (*re*).

TD Group: Aloud Reading	Estimate	SE	t Value	Pr (>|t|)
intercept (cont)	0.24	0.02	14.60	<$2\times 10^{-16}$
position (cont)	−0.01	0.01	−8.69	<$2\times 10^{-16}$
class (cont)	−0.04	0.01	−7.28	<0.001
position (func)	−0.01	0.01	−4.67	<0.001
class (func)	−0.01	0.01	−0.22	0.83
participants (re)	<$2\times 10^{-16}$			
word tokens (re)	<$2\times 10^{-16}$		R^2^	14.6%
**TD Group: Silent Reading**	**Estimate**	**SE**	**t Value**	**Pr(>|t|)**
intercept (cont)	0.22	0.01	15.49	<$2\times 10^{-16}$
position (cont)	−0.01	0.01	−9.78	<$2\times 10^{-16}$
class (cont)	−0.04	0.01	−7.81	<0.001
position (func)	−0.02	0.01	−7.96	<0.001
class (func)	−0.01	0.01	−2.49	<0.05
participants (re)	<$2\times 10^{-16}$			
word tokens (re)	<$2\times 10^{-16}$		R^2^	16.7%

**Table 8 brainsci-12-01316-t008:** *GAM* coefficients for models fitting tracking time on symbols as a function of symbol position in the word (*position*) in interaction with content vs. function words (*cont, func*) and grade level (*class*) for the two reading modalities. Participants in the ASD group and word tokens are added as random effects (*re*).

ASD Group: Aloud Reading	Estimate	SE	t Value	Pr (>|t|)
intercept (cont)	0.23	0.03	8.09	<$2\times 10^{-16}$
position (cont)	−0.01	0.01	−6.19	<0.001
class (cont)	−0.03	0.01	−3.29	0.001
position (func)	−0.01	0.01	−1.63	0.1
class (func)	0.01	0.01	0.14	0.89
participants (re)	<$2\times 10^{-16}$			
word tokens (re)	<0.001		R^2^	8.67%
**ASD Group: Silent Reading**	**Estimate**	**SE**	**t Value**	**Pr(>|t|)**
intercept (cont)	0.1	0.01	2.22	<0.005
position (cont)	−0.01	0.01	−1.70	0.09
class (cont)	−0.01	0.01	−0.90	0.37
position (func)	−0.01	0.01	−0.89	0.37
class (func)	0.01	0.01	0.07	0.94
participants (re)	<$2\times 10^{-16}$			
word tokens (re)	<0.005		R^2^	4.08%

**Table 9 brainsci-12-01316-t009:** Mean per-word tracking time (in seconds), and standard deviation, for 3 ranges of comprehension accuracy (low accuracy = 1st quartile; medium accuracy = 2nd, 3rd quartiles; high accuracy = 4th quartile), aggregated by reading modalities for ASD and TD children.

ASD Group	Mean	St.Deviation
low accuracy	0.67	0.94
medium accuracy	0.59	0.72
high accuracy	0.48	0.70
**TD Group**	**Mean**	**St.Deviation**
low accuracy	0.52	0.67
medium accuracy	0.49	0.61
high accuracy	0.30	0.29

## Data Availability

Data collected and analyzed within the current study cannot be made publicly available. Aggregated anonymized data may be requested to authors for academic purposes.

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
