# Peer review of "Reading Behaviors through Patterns of Finger-Tracking in Italian Children with Autism Spectrum Disorder"

_brainsci, 2022, doi:10.3390/brainsci12101316_

Round 1
Reviewer 1 Report
This study explores the finger movement of children with ASD during reading activities. The study provide insights into possible perceptual functioning of individuals with ASD. At the same time, the authors proposed tracking method based on an ecological application, which might be useful in both early diagnosis and treatment.
I have some major concerns about the study.
1. The ASD and TD groups are different in terms of chronological age, gender ratio, verbal IQ, and non-verbal IQ, even though the authors mentioned that they are matched on age and grade level.
2. The sample size is very small especially when the authors tried to conduct cross-sectional analyses. How many children are in each grade level (grade2 to 5)? Without knowing more information of the participants, it is hard to make sense of the results, especially from a developmental perspective.
3. Children with ASD demonstrated more variability regarding the accuracy of comprehension in the silent reading condition. This finding deserves more explanation
Other minor issues include:
1. 1/100 according to the Centers for Disease Control and Prevention (CDC) (p.1, line 32-33). Please cite the most up-to-date figure, if you decide to cite the data from the US. Also please provide the prevalence rate of ASD in Italy.
2. In contrast with previous literature showing that ASD readers are good at reading words, 67 but not as good at reading non-words [16,17] (p.2, line 66-67). There is an extra “tab” before “but”, please check typesetting issues throughout the manuscript.
3. (p=0.38) (p. 10, line 239). There should be a space before and after the equal sign. Please check spacing issues throughout the manuscript.
Author Response
We do thank the Reviewer 1 for the precious comments and suggestions. We revised/integrated the paper accordingly. Detailed replies (in plain text) follow each single issue (reported in italics).
The ASD and TD groups are different in terms of chronological age, gender ratio, verbal IQ, and non-verbal IQ, even though the authors mentioned that they are matched on age and grade level. 2. The sample size is very small especially when the authors tried to conduct cross-sectional analyses. How many children are in each grade level (grade2 to 5)? Without knowing more information of the participants, it is hard to make sense of the results, especially from a developmental perspective. --> we added these figures in Table 1 and 2.
Children with ASD demonstrated more variability regarding the accuracy of comprehension in the silent reading condition. This finding deserves more explanation. --> we added more considerations on this issue.
1/100 according to the Centers for Disease Control and Prevention (CDC) (p.1, line 32-33). Please cite the most up-to-date figure, if you decide to cite the data from the US. Also please provide the prevalence rate of ASD in Italy. --> we updated figures with both US and Italian ones.
In contrast with previous literature showing that ASD readers are good at reading words, 67 but not as good at reading non-words [16,17] (p.2, line 66-67). There is an extra “tab” before “but”, please check typesetting issues throughout the manuscript. --> the originally submitted paper is a pdf generated from latex source, thus without inappropriate spacing. The extra tab is only in the converted Word version.
(p=0.38) (p. 10, line 239). There should be a space before and after the equal sign. Please check spacing issues throughout the manuscript. --> spaced
Reviewer 2 Report
The paper aims to propose a protocol for large-scale collection of reading data in high-functioning Autism Spectrum Disorder (ASD) children, based on recording the finger movements of a subject reading a text displayed on a tablet touchscreen. The authors suggest that the use of a tablet as a tactile interface for visual perception analysis can offer a robust experimental protocol for large scale data screening in ASD research and clinical settings.
Nevertheless available information is not enough to present experimental protocol. Information about text complexity and length, procedure and instructions, list of variables, number of observations, descriptive statistics etc. are necessary.
There are many doubts for sample composition. Autism is characterized by high variability of speech and cognitive development, such parameters can significantly influence the results. Nevertheless there is no data regarding IQ and speech scores in experimental group. Groups were matched only on age and school grade. It is particularly important to match groups on language development considering inclusion of words length and frequency as independent variable in regression models.
Considering experimental group size (20 subjects), division of groups according to the grade raises doubts about representativeness of the results.
It would be well to provide regression model parameters, p-values on graphs, as well as show distribution of data together with regression approximation line.
Summing up, provided information is not enough for understanding experimental protocol in details neither for presenting reading skills development pattern in ASD.
Author Response
We do thank the Reviewer 1 for the precious comments and suggestions. We revised/integrated the paper accordingly. Detailed replies (in plain text) follow each single issue (reported in italics).
Information about text complexity and length, procedure and instructions, list of variables, number of observations, descriptive statistics etc. are necessary. --> we added these figures in Table 3
There are many doubts for sample composition. Autism is characterized by high variability of speech and cognitive development, such parameters can significantly influence the results. Nevertheless there is no data regarding IQ and speech scores in experimental group. --> we added these figures in Table 1
Groups were matched only on age and school grade. It is particularly important to match groups on language development considering inclusion of words length and frequency as independent variable in regression models. Considering experimental group size (20 subjects), division of groups according to the grade raises doubts about representativeness of the results. --> we added detailed figures in Table 1 and 2. We are well aware that it is a very preliminary study; however, results are encouraging since they show the feasibility of such an approach.
It would be well to provide regression model parameters, p-values on graphs, as well as show distribution of data together with regression approximation line. --> Table 4 provides model parameters, p-values and model R-squared for Figure 2; Table 5 for Figure 3; Table 6 for Figure 4; Table 7 for Figure 5; Table 8 for Figure 6. We also added a direct comparison between groups in section 3.1
Reviewer 3 Report
In this paper, the authors describe reading behavior in children with ASD and a control group of typically developing children using a finger-tracking-during-reading paradigm. In addition to the analysis of tracking time during silent and oral reading between the groups, the authors estimate the effects of word length and frequency across development, the effect of word class (content vs. function words), as well as tracking time during reading words followed by punctuation marks in both groups of participants.
This is a nice and important study that uses a novel paradigm to assess reading in a large group of children with ASD. However, there are some issues that should be addressed or clarified.
Major issues
1. Introduction
1.1. Reading patterns in languages may depend on the orthographic system of a language (e.g., see Liversedge et al., 2016, Siegelman et al., 2022; Schroeder et al., 2021). The authors should be more precise when reviewing the literature about finger movements and reading effects (lines 49-50): what languages and scripts were compared, and what effects were replicated?
1.2. The beginning of the third paragraph of the Introduction (lines 56-81) is confusing. Did the authors assess “three complementary aspects of reading behavior” by means of behavioral testing in addition to the finger-tracking paradigm or did they focus on different aspects of their reading dataset?
1.3. I was confused by the motivation for the analysis (i). The authors hypothesized that children with ASD should read words slower than TD children, and at the same time, they should read words faster than non-words (lines 67-68). However, in the Results section, the authors compared the reading of content and function words in children with ASD and TD in separate models. It does not allow them to compare the effects between the groups. Non-words were not included in the experiment at all. The motivation (i) should be rephrased according to the actual analysis presented in the paper.
1.4. The authors should add motivation to their analysis of word length and frequency effects in the two groups of participants across grades.
2. The Materials and Methods section should be restructured and expanded with a description of the materials and data analysis.
2.1. The description of participants should include means and SD for age in each group. If the authors did not match the groups by age and grade on a one-on-one basis, they should also include the information about the analysis (e.g., t-test) showing that the groups did not differ.
Did the authors check that children with ASD had normal non-verbal IQ?
2.2. The description of the reading materials does not allow the readers to follow the results. The authors should add a separate sub-section “Materials” and provide there the information about the text, e.g., topic, text length, mean word length and frequency, number of content and function words (as well as the information in lines 218-221).
How many comprehension questions followed the text? Could the authors provide examples of comprehension questions?
2.3. In the Data analysis section the authors should provide the rationale for using Generalized Additive Models over linear mixed-effects models.
Did the authors transform (e.g., log-transformation, centering, etc.) their predictors? How were the categorical variables (group, modality) contrast-coded?
See Schad et al. (2020). How to capitalize on a priori contrasts in linear (mixed) models: A tutorial. https://doi.org/10.1016/j.jml.2019.104038
Also, the Data analysis section would benefit from adding a data analysis plan. That will help the readers follow the Results section.
3. Results
3.1. Based on the description of the analyses in the Results section, I conclude that the authors built separate models for ASD and TD groups and then described the effects and compared these descriptions. However, it does not allow us to prove any difference between the groups. I suggest the authors rerun the analysis and include group as an independent variable in all the models.
3.2. In Tables 2 and 3 the authors present the results of GAMs for the interaction of length/frequency and grade. To me, the comparison of the effect of word length in the silent reading condition in grade 5 with the effect of length in the oral reading condition in grade 2 makes no sense.
I suggest the authors re-code reading modality with sum contrasts (Schad et al., 2020) and add grade as a continuous variable. To analyze significant interaction effects, the authors might want to use the emtrends function in the R package emmeans (Lenth et al., 2021) which contrasts slopes of the predictors in the model.
3.3. In section 3.2. Word internal position, the authors compared tracking times for content and function words. Function words are usually shorter and more frequent than content words. However, the authors do not control for length and frequency in their models. It is possible that the faster reading of function words (lines 252-253) was due to their length and frequency and not word class.
3.4. I think there might be some confusion in the description of the results in lines 268-270 (17.28% of untracked word tokens in ASD group in silent condition vs. 57.21% in oral condition – why “especially” refers to silent reading condition?).
4. Discussion
To facilitate readers’ comprehension of their findings, the authors should align the Results and the Discussion sections. The latter starts with the discussion of reading comprehension which is the penultimate topic of the Results.
Minor issues
5. aloud reading → oral reading
6. Please round very low p-values (e.g. 5.84-07, etc.) to p<0.001, e.g. Figure 1, Table 1, etc.
7. Figure’s axes and legends are not aligned (see Figure 2, Figure 7)
8. The first column heading in Tables 4 and 5 is repeated twice
9. The group of typically developing children is named inconsistently in the text (e.g., “controls” on line 52, “control children” in Figure 3, “TD group” in Table 2).
10. st.err → SE (in Tables)
11. line 46: developmental readers → developing readers
12. lines 123-124: please add a reference to Readlet project
13. line 178: progressive reduction → decrease
14. line 181: later readers → older readers
15. line 278: ensuing word → following word
16. line 303: please add a reference.
Author Response
We do thank the Reviewer 3 for the precious comments and suggestions. We revised/integrated the paper accordingly. Detailed replies (in plain text) follow each single issue (reported in italics).
Reading patterns in languages may depend on the orthographic system of a language (e.g., see Liversedge et al., 2016, Siegelman et al., 2022; Schroeder et al., 2021). The authors should be more precise when reviewing the literature about finger movements and reading effects (lines 49-50): what languages and scripts were compared, and what effects were replicated? --> A short paragraph was added describing the reviewed evidence in relation to Italian transparent orthography and the specific effects that were replicated in that context.
The beginning of the third paragraph of the Introduction (lines 56-81) is confusing. Did the authors assess “three complementary aspects of reading behavior” by means of behavioral testing in addition to the finger-tracking paradigm or did they focus on different aspects of their reading dataset? --> We wrote a new paragraph clarifying that we intended to focus on different aspects of our tracking dataset and why.
I was confused by the motivation for the analysis (i). The authors hypothesized that children with ASD should read words slower than TD children, and at the same time, they should read words faster than non-words (lines 67-68). However, in the Results section, the authors compared the reading of content and function words in children with ASD and TD in separate models. It does not allow them to compare the effects between the groups. Non-words were not included in the experiment at all. The motivation (i) should be rephrased according to the actual analysis presented in the paper. --> Our writing was in fact a bit misleading. We intended to focus on lexical vs. sublexical effects, which obviously have a strong connection with word vs. nonword reading but were in fact explored and investigated in our analysis through frequency and length effects. We therefore dropped any explicit reference to nonword reading, which is not tested experimentally.
The authors should add motivation to their analysis of word length and frequency effects in the two groups of participants across grades. --> we added a short paragraph making a direct connection with the background motivations outlined in the introduction.
The Materials and Methods section should be restructured and expanded with a description of the materials and data analysis. --> “Materials” subsection has been added
The description of participants should include means and SD for age in each group. If the authors did not match the groups by age and grade on a one-on-one basis, they should also include the information about the analysis (e.g., t-test) showing that the groups did not differ. --> we added more detailed information about the distribution per group by grade level.
Did the authors check that children with ASD had normal non-verbal IQ? --> we added this figure
The description of the reading materials does not allow the readers to follow the results. The authors should add a separate sub-section “Materials” and provide there the information about the text, e.g., topic, text length, mean word length and frequency, number of content and function words (as well as the information in lines 218-221). How many comprehension questions followed the text? Could the authors provide examples of comprehension questions? --> a “Materials” subsection has been added with detailed information on these issues
In the Data analysis section the authors should provide the rationale for using Generalized Additive Models over linear mixed-effects models. --> we added a footnote (see ft 1).
Did the authors transform (e.g., log-transformation, centering, etc.) their predictors? How were the categorical variables (group, modality) contrast-coded? --> Only corpus-based word frequencies were log-transformed.
Based on the description of the analyses in the Results section, I conclude that the authors built separate models for ASD and TD groups and then described the effects and compared these descriptions. However, it does not allow us to prove any difference between the groups. I suggest the authors rerun the analysis and include group as an independent variable in all the models. --> we already included such a direct comparison model, as originally reported in footnote 1. We moved it to section 3.1 and added more details (including a regression plot, see Figure 5)
In Tables 2 and 3 the authors present the results of GAMs for the interaction of length/frequency and grade. To me, the comparison of the effect of word length in the silent reading condition in grade 5 with the effect of length in the oral reading condition in grade 2 makes no sense. I suggest the authors re-code reading modality with sum contrasts (Schad et al., 2020) and add grade as a continuous variable. To analyze significant interaction effects, the authors might want to use the emtrends function in the R package emmeans (Lenth et al., 2021) which contrasts slopes of the predictors in the model. --> We checked for grade level as continuous variable and results are consistent; we simply reported in Tables the models related to the Figures.
In section 3.2. Word internal position, the authors compared tracking times for content and function words. Function words are usually shorter and more frequent than content words. However, the authors do not control for length and frequency in their models. It is possible that the faster reading of function words (lines 252-253) was due to their length and frequency and not word class. --> we measured tracking time per symbol and not per word: thus plots and models are referred to the tendency of accelerating while reading words, which - on average - is different for content and function words, and - importantly for our concerns - different for TD and ASD.
I think there might be some confusion in the description of the results in lines 268-270 (17.28% of untracked word tokens in ASD group in silent condition vs. 57.21% in oral condition – why “especially” refers to silent reading condition?). --> the original text is: This is particularly evident in the silent reading condition, where we suspect that some ASD children might have not read the text carefully enough. In fact, we found more untracked word tokens in the ASD than in the TD group: 17.28% in ASD as compared to 14.38% in TD, especially in silent reading (57.21%, with 42.79% in aloud reading). Thus the explanation precedes. We added some comments on this issue.
aloud reading → oral reading à since the two words are synonyms, we prefer “aloud reading”
Please round very low p-values (e.g. 5.84-07, etc.) to p<0.001, e.g. Figure 1, Table 1, etc. --> rounded
Figure’s axes and legends are not aligned (see Figure 2, Figure 7) -->aligned
The first column heading in Tables 4 and 5 is repeated twice --> yes, since there are two models each related to aloud and silent reading
The group of typically developing children is named inconsistently in the text (e.g., “controls” on line 52, “control children” in Figure 3, “TD group” in Table 2). --> made consistent in the text and plots
st.err → SE (in Tables) --> simplified in its acronym
line 46: developmental readers → developing readers --> thanks for spotting this out: we corrected all occurrences of it
lines 123-124: please add a reference to Readlet project --> we added reference to the project website. In addition, a full reference can be found in the Funding section, at the end of the paper
line 178: progressive reduction → decrease --> changed
line 181: later readers → older readers --> changed
line 278: ensuing word → following word --> since we refer to words in a connected text, we prefer “ensuing word
line 303: please add a reference. --> a new, specific reference was added
Round 2
Reviewer 1 Report
The authors addressed my previous concerns and the manuscript has been improved.